# Who Makes or Breaks Energy Policymaking in the Caribbean Small Island Jurisdictions? A Study of Stakeholders' Perceptions

Xiaoyu Liu [1,*] , Jahan Ara Peerally [2], Claudia De Fuentes [1], David Ince [3] and Harrie Vredenburg [3]

1   Sobey School of Business, Saint Mary's University, Halifax, NS B3H 3C3, Canada; claudia.deFuentes@smu.ca
2   Department of International Business, HEC Montréal, Montréal, QC H3T 2A7, Canada; jahan.peerally@hec.ca
3   Haskayne School of Business, University of Calgary, Calgary, AB T2N 1N4, Canada; incephd@gmail.com (D.I.); harrie.vredenburg@haskayne.ucalgary.ca (H.V.)
*   Correspondence: xiaoyu.liu@smu.ca

**Abstract:** While most studies view small island economies as a homogenous group with multiple similar vulnerabilities, few studies argue that they are a heterogenous group due to their political jurisdictions (independent versus dependent economies), with mostly environmental vulnerabilities in common. Departing from these two premises, our study is the first empirical attempt at examining inter-small island jurisdiction (SIJ) heterogeneity from the social construct perspective of stakeholders' perceptions and within the context of environmental sustainability and energy policymaking. We quantitatively explore, across 34 Caribbean SIJs, multiple stakeholders' perceptions of the influence of the electricity sector as a leader in environmental performance. The results show that when the governments of independent SIJs exclude electricity sector stakeholders and include other primary energy stakeholders in energy policymaking, the electricity sector actors are better perceived as leaders in environmental performance. In a global context where inclusiveness is important for sustainability, this finding suggests that within the systemic contexts of SIJs, stakeholders view the exclusion of powerful incumbent energy actors from policymaking as a viable approach for moving the environmental sustainability mandate forward. Our study has implications for policymakers and scholars on the democratic process of policymaking, and for practitioners in terms of building social trust.

**Keywords:** Caribbean SIJs; developing economies; electricity sector; energy policymaking; environmental sustainability; stakeholders' perceptions

## 1. Introduction

More than 30 years ago, the United Nations [1] stated that "a safe and sustainable energy pathway is crucial to sustainable development; we have not yet found it". Since then, international development organizations and advanced economies have provided aid and support to developing economies for the implementation of sustainable development policies, practices, and technologies [2]. A group of such recipient economies are the small island developing states (SIDS) [2,3].

SIDS are often presented as facing an array of uniquely similar developmental and sustainability challenges arising from their remote locations, poor economic diversification, vulnerability to climatic fluctuations, and great outward economic dependence on an increasingly unstable and globalized world [4]. However, a sparse literature [5–7] conceives these islands as being heterogenous in terms of their political jurisdictions and refers to them as small island jurisdictions (SIJs). SIJs are either independent political jurisdictions or dependent, non-self-governing, non-sovereign island economies.

The SIJ literature emphasizes that dependent islands perform better than independent ones across various factors. Baldacchino [5] conceptually argues that dependent SIJs are more innovative in their development strategies than politically independent SIJs, and calls for further "timely investigation" into the sustainable development approaches that

result from the differences between the SIJs' political jurisdictions [5]. Additionally, some studies have shown that dependent SIJs tend to be richer per capita than independent ones [8,9]. It, therefore, follows that dependent SIJs which are intricately tied to more advanced economies, with wider and deeper resources, benefit from additional mainland and institutional support in their efforts towards sustainable development, when compared to independent SIJs.

The extant literature on these economies and their energy sector treats them as a homogenous group and concurs on the following points. First, that the Caribbean is the second most environmentally hazard-prone region in the world where natural disasters, along with climate change, loss of biodiversity, depleted freshwater and pollution are the main environmental challenges [3,10]. Second, that the Caribbean SIJs have made little progress in terms of improving the role of renewables in their energy services in recent decades [11]. Third, such lack of progress is believed to be further hampered by the considerably increasing demand for energy in the Caribbean over the past decade, and by the fact that large deposits of high-grade oil have recently been found off the coast of Guyana, while Grenada has found oil and gas in huge commercial quantities [12]. Fourth, OECD 10 reports that while most Caribbean SIJs aim to improve the role of renewables, hindering issues include fiscal constraints, data gaps, lack of local capabilities, weak local markets, incomplete or inadequate governance frameworks, and weak enforcement of regulations.

One of the first departures from these generalizations across Caribbean SIJs and the energy sector, includes the work of Shirley and Kammen [13] who infer that "(n)uances between Caribbean SIDS' needs and contexts" may be part of the reason that a cohesive regional energy policy is difficult to develop. However, these nuances at the SIJ level have not been empirically explored, especially from a social construct perspective based on stakeholders' perceptions and within the context of environmental sustainability and energy policymaking. Our study is a first attempt at addressing this empirical gap within the existing literature.

Our study aims to explore this notion of nuances or heterogeneity across a group of SIJs which are geographically close, but have experienced a temporally different form and quality of colonialism, with arguably different resource endowments and systems of regulation and governance. It examines the stakeholders' perceptions of the electricity sector as a leader in environmental performance and the impact of governments and their involvement of primary energy stakeholders in energy policymaking, and the role of international development organizations in the electricity sector. The study is of key relevance since sound energy policymaking is a strong basis for an environmentally sustainable electricity sector as well as for a holistic sustainable development strategy, which have been the overarching priority of Caribbean SIJs for decades. The stakeholders involved in addressing the SIJs' environmental sustainability and energy policymaking include: (1) governments (national independent, local territorial governments); (2) primary energy stakeholders in the electricity sector (electricity producers, electricity utility providers); (3) primary energy stakeholders in the non-electricity sector (traditional oil and gas companies, renewable energy companies); (4) primary non-energy stakeholders (manufacturing industries, local agencies of international development organizations); and (5) secondary non-energy stakeholders (domestic and international NGOs and domestic research organizations including universities).

We focus on the above stakeholders' perceptions of two leading actors, namely the local governments and international development organizations, in the context of environmental sustainability, and energy policies for the electricity sector. Our research questions are:

(i)  Are local governments and international development organizations, in the Caribbean SIJs, perceived by key stakeholders as significantly impacting the positioning of the electricity sector as a leader in environmental performance?

(ii) Is there a difference in the stakeholders' perceptions based on the Caribbean island's political jurisdiction?

We propose two sets of hypotheses to address these two research questions. First, we argue that there are positive relationships between stakeholders' perceptions of the positioning of the electricity sector as a leader in environmental performance and the involvement of governments and international organizations (Hypotheses 1a and 1b). Second, we argue that the political jurisdiction of the Caribbean islands moderates these relationships (Hypotheses 2a and 2b).

The remainder of the paper is organized as follows. In Section 2, we present the conceptual framework of our study and conduct a tailored literature review on the pertinent elements of the framework; we also present the methods used in the study. We present our findings in Section 3. In Section 4, we discuss the implications of our findings; and in Section 5, we provide some concluding remarks.

## 2. Materials and Methods

### 2.1. Conceptual Framework: Key Actors and Stakeholders' Perceptions of Sustainable Energy Production

We developed a conceptual framework (Figure 1) which provides an overview of the theoretical structure for our hypotheses. The literature review covers the studies which are relevant to our research, with special focus on developing economies and the Caribbean context. First, we discuss the relevance of stakeholders' perceptions and stakeholder involvement for sustainable development, which form the foundation for our analysis. Second, we elaborate on the role of governments and international development organizations in achieving sustainability. Third, we discuss the emerging literature on the often-conflicting roles of incumbent stakeholders in the development of sustainable energy sectors.

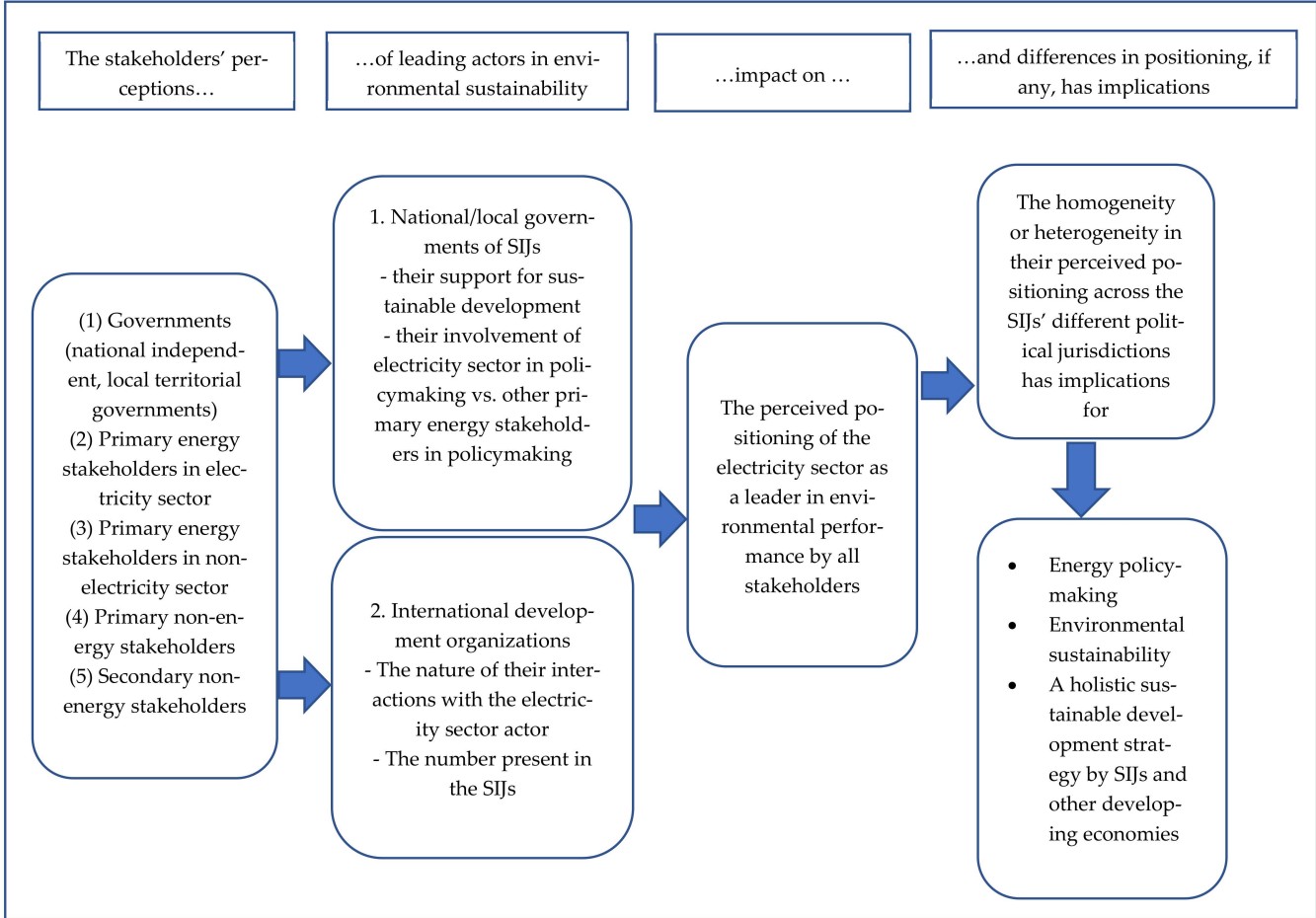

**Figure 1.** Conceptual framework.

### 2.1.1. Stakeholders' Perceptions, Stakeholder Involvement, and Implications for Sustainable Development

Complex and often ambiguous concepts such as sustainability are subject to various perceptions and interpretations [14]. However, the achievement of common ground in stakeholder perceptions can influence business strategies and actions, and usually leads to behavioral changes [15]. Therefore, searching for common ground between these differing perceptions and interpretations among a broad range of stakeholder groups is important [14], as a better understanding of stakeholder perception and its influence on decision-making can help in better aligning sustainability policies and programs with important stakeholders' interests and capabilities [15].

In business practices, cultural, institutional, and psychological barriers inhibit the achievement of common ground between diverging stakeholder perceptions, hence leading to biases between these same stakeholders [15]. Stakeholder involvement, participation, and collaboration are therefore significant ingredients for reshaping their actions and, in turn, play a critical role in building environmental sustainability [16–18]. For example, at the firm level, both primary and secondary stakeholders possess the necessary knowledge which when shared, enables them to better understand their organizational environment [17]. At the industry level, a collaborative culture with different stakeholders facilitates the development of sustainable industries [19]. At the government level, including stakeholders in policymaking processes provides marginal groups with the opportunity to empower their voices, enhances the democratic nature of the policymaking process, and increases the mutual information exchanges between stakeholders and policymakers [18]. Therefore, emerging studies in sustainability increasingly use data on stakeholder involvement in energy policymaking.

Matsuo and Schmidt [20] used data from Mexico and South Africa on the involvement of local, foreign, public, and private stakeholders, and they show that these two countries' prioritization of the trade-off between traditional input-intensive industrialization and low-carbon energy technologies led to a divergence of their renewable energy outcomes and that this will undoubtedly impact future developments of cleaner forms of energy production. Additionally, Patala et al. [21], using investment data by multinational energy utility enterprises, empirically show that a systemic and coordinated collaboration between multiple stakeholders, such as national governments, international entities (including both international NGOs and multinational enterprises), supports and assists energy transition to low-carbon energies.

Moreover, stakeholder involvement can improve the decision-making process by integrating additional information, new ideas and stakeholders' knowledge into the process, thereby increasing the likelihood of high-quality decisions [22,23]. Stakeholder involvement in addressing various sustainability issues acts as a preventative mechanism against maladaptation (defined as a practice that increases vulnerability by Hopkins [24]) in the implementation of policies and strategies for sustainability. Furthermore, the development of a sustainable energy and electricity sector requires transformations in terms of energy sources, operations, and technologies, but also a deep transformation of incumbent stakeholders involved in this developmental process. Therefore, examining the Caribbean SIJs' and other developing economies' stakeholder perceptions of different leading actors in sustainable development and the electricity sector is important for scholars, practitioners, and policymakers.

### 2.1.2. Governments and International Development Organizations as Key Actors for Achieving Sustainability

The innovation studies literature discusses the importance of governments in developing innovations that tackle grand challenges including sustainable energy sources. Mazzucato [25] and Mazzucato and Semieniuk [26], for example, elaborate on the importance of the government as an active actor, and not only as an enabler, for the development and implementation of such innovations. Without a systemic active approach which brings

together all key stakeholders within economies, it would not have been possible to develop and implement radical or disruptive innovations such as the internet and satellites [25].

Discussions within the innovation studies literature, based on the systems of innovation approach [27] and innovation policies [28], have also highlighted the proactive role of governments and international development organizations in the emergence of innovations for achieving national objectives. The engagement for building the capacities and capabilities needed for implementing sustainable development requires a fundamental transformation of sociotechnical systems [29,30]. Similarly, the creation of capacities for sustainable production of energy and other grand challenges requires a systemic perspective where the government plays a proactive role in collaboration with other key local and foreign stakeholders. Thus, the transition from non-renewable to renewable energy sources requires the engagement of several stakeholders as key actors wherein the active role of the government is not only as regulator, but also as an enabler for the development of the strategies, technologies, and capacities essential for such a transition.

For developing economies, these strategies, technologies, and capacities are usually acquired from more advanced foreign locations, where international development organizations and local governments play important roles as orchestrators for their transfer. Thus, the systemic perspective for tackling grand challenges axiomatically implies the involvement of both local and foreign actors including international development organizations that are often well positioned to provide the necessary support and contribution in terms of financial resources, building capacity, and mobilization of stakeholders. For example, Ince et al. [31] underline that international organizations play a prominent role in advising Caribbean SIJs on energy policy and have an impact on their use of renewable energy technology. Thus, the involvement and support of international development organizations for the development of sustainable energies in developing economies is vital.

To conclude, we view local governments and international development organizations, when compared to the other stakeholders, as the leading actors in implementing, and even enforcing, sustainable energy policies due to their legitimacy and their unparalleled access to funds, technical expertise, and networks of stakeholders in building capacity and capabilities for implementing a holistic sustainable development strategy. Therefore, we develop our first set of hypotheses:

**Hypothesis 1 (H1).** *There is a positive relationship between stakeholders' perceptions of the positioning of the electricity sector as a leader in environmental performance and:*
*(H1a) the involvement of governments.*
*(H1b) international organizations.*

2.1.3. Conflicts of Interest between Stakeholders, Energy Policymaking, and Political Jurisdictions

It is important to implement and achieve a sustainable outcome in the energy sector's capacity building process [32,33]. Rapid and deep reductions in greenhouse gas emissions are needed, through global collaborative efforts such as the Paris Agreement, to avoid dangerous climate change. However, developing sustainable strategies to preserve the planet is both an important and challenging endeavor that requires collaboration among different stakeholder groups [34]. Regarding the Caribbean region specifically, most of the energy production capacity in the Caribbean is derived from non-renewable sources. Around 87% of primary energy consumed is in the form of petroleum, while less than 13% is produced using renewable sources including wind, hydro, and thermal [35]. One explanation for this gap between the use of non-renewable versus renewable sources of energy within the Caribbean can be derived from Ince et al. [31], who state that in many cases, " . . . the electricity utility itself is interested in becoming a player in the renewable energy market and it requires a degree of skill to be able to manage the conflicting goals of the utility in ensuring that safety, reliability, and accessibility to power is maintained while ensuring that the utility also has the opportunity to fairly compete in the emerging and innovative field of renewable energy", and "the incumbent utility has taken the lead and

been able to ensure that renewable energy development has taken place while maintaining a strategically important position as a major player in the new sector" (pp. 708). Hence, the development of renewable energy presents continuous challenges in terms of infrastructure, capacity, and the implementation of policies and actions for capacity building [31].

Since some incumbent actors (e.g., the electricity sector) have an influential hold on the development of sustainable energy sectors, it is crucial to have a deeper understanding of different stakeholders' perceptions of the electricity sector as a leader in environmental performance and its implications for energy policymaking, environmental sustainability, and the implementation and achievement of a holistic sustainable development strategy. Additionally, studies on energy and innovation highlight the centrality and coherence of policy design for steering policy outcomes [20,36–38]. For example, industrial policy measures can enable the accumulation of firm-level capabilities that are critical for developing economies seeking to escape the middle-income trap [39]. Good science, technology, and innovation policies and instruments can provide in-depth analyses on how low-carbon energy policy influences the emergence of local green industries [20,28]. Furthermore, stakeholders involved in different political settings have a different response pace with regards to environment and energy policies [36]. As a result, the different response pace might lead to conflicts of interest among stakeholders. As indicated by Ince et al. [31], there are conflicts of interest between the incumbent electricity sector and independent renewable energy producers in the Caribbean SIJs. It is thus necessary to design effective policies to reconcile the conflicting interests of incumbent coal and oil companies with those of the emerging low-carbon sectors and technologies [40].

On the one hand, the conflicting interests between the incumbent stakeholders and those who focus on renewable and sustainable energy sources require the implementation of policies which support the development of technological capacity [40]. On the other hand, the conflicting interests require the integration of "macro-level" considerations (i.e., which include governance elements), with "micro-level" ones, such as on-the-ground calibrations of policy objectives and instruments during policymaking processes [14,41].

Building on the above discussion, our study aims to add to this emerging literature on the conflicts of interest between incumbent energy stakeholders and other stakeholders' perceptions of this dynamism, and explore the implications for energy policymaking and environmental sustainability of SIJs. It also quantitatively explores the notion of nuances or heterogeneity in a group of SIJs that have been so far examined as a homogenous group within the context of energy policymaking. Political jurisdictions of Caribbean SIJs include dependent SIJS and independent SIJs. Comparing governments of independent and dependent SIJs is conceptually sound since, as highlighted in several studies [31], the latter have their own local territorial governments that are locally elected and possess a level of autonomy that confers policy and decision-making control onto them. Economically speaking, dependent SIJs are intricately tied to more advanced economies, with wider and deeper resources, and they benefit from additional mainland and institutional support in their efforts towards sustainable development, when compared to independent SIJs. Therefore, it has been argued that some dependent SIJs perform economically better than some larger continental states [42,43], and dependent SIJs tend to be richer per capita than independent ones [8,9].

We argue that these differences in economic and political aspects between dependent and independent SIJs will influence our proposed relationships between stakeholders' perceptions on the positioning of the electricity sector as a leader in environmental performance and the involvement of governments and international development organizations. Therefore, we develop our second set of hypotheses as follows:

**Hypothesis 2 (H2).** *The Caribbean islands' political jurisdiction moderates the relationship:*

*(H2a). between stakeholders' perceptions of the positioning of the electricity sector as a leader in environmental performance and the involvement of governments.*

*(H2b). between stakeholders' perceptions of the positioning of the electricity sector as a leader in environmental performance and international development organizations.*

*2.2. Methods*

2.2.1. Research Background and Data Collection

This study examines stakeholders' perceptions of the positioning of the electricity sector as a leader in environmental performance as impacted by governments and their involvement of primary energy stakeholders in policymaking, and international development organizations. The different Caribbean political jurisdictions covered in our study and their relevant indicators namely, geographical area, cost of electricity, population size, and political status, are listed in Table 1. As seen in Table 1, these indicators are heteregenous across Caribbean SIJs. This study aims to examine this heterogeneity across these islands specifically in relation to their political jurisdictions.

A survey questionnaire was designed to explore the Caribbean SIJs stakeholders' perspectives on the local governments' (national independent versus local territorial governments) and international development organizations impact on the electricity sector's positioning within the local business environment as a leader in environmental performance. Invitations were sent out to potential respondents via email with a link to an online Qualtrics survey. The targeted respondents included a diverse group of stakeholders listed in Figure 1 and which are also presented in Table 2. To facilitate respondents' participation and to accommodate the different Caribbean SIJs' languages, the surveys were prepared in English, French, Spanish, and Dutch. A total of 535 surveys were sent out with a 26% response rate representing 141 usable questionnaires. The questionnaires were sent out to all 36 Caribbean SIJs, however, there were no responses from St. Martin and St. Barthelemy, which therefore brings our SIJ population to 34. We tested for demographic-based non-response bias and there was no evidence of systematic bias.

2.2.2. Measurement of Variables

In this subsection, we describe the dependent and independent variables which were derived from our survey questions. The survey questions related to these variables are detailed in Appendix A.

- Dependent variable. The dependent variable measures the positioning of the electricity sector as a leader in terms of environmental performance within the national business environment. This variable is based on multiple stakeholders' perceptions. In Table 2, we present the descriptive statistics of the different stakeholder groups and their aggregated rated responses on the perceived positioning of the electricity sector as a leader in environmental performance.
- Independent variables. The study's five independent variables relate to local governments and international development organizations. They are explained hereunder.

(1) International development organizations: Number of interactions. We implemented two independent variables as a measure of the role of international development organizations in the Caribbean SIJs' energy sector. The first variable is based on the number of times each stakeholder group interacted with each of the international development organizations (international and regional) present in the 34 Caribbean SIJs. All international development organizations present in the Caribbean SIJs were previously identified by the researchers and inserted in the questionnaire and are listed in Appendix A.

(2) International development organizations: Attitudes. The second variable is derived from factor analysis. We designed seven questions to measure the nature of the responding stakeholders' interactions with these previously identified international development organizations. Through an exploratory factor analysis of their answers, we identified one factor which is the perceived international development organiza-

tions' attitudes and interests toward the electricity sector and the SIJs. The specific factor loadings are shown in Table 3.

**Table 1.** List of political jurisdictions and their key indicators.

| Caribbean Economies | Area/km$^2$ | Population | Cost of Electricity (cents/kWh) | Political Status | Political Jurisdictions * |
|---|---|---|---|---|---|
| Anguilla | 90 | 13,500 | 35.71 | British Overseas Territory | 0 |
| Antigua and Barbuda | 442 | 89,000 | 38.01 | British Colony | 1 |
| Aruba | 180 | 108,000 | 25.24 | Member of the Kingdom of the Netherlands | 0 |
| Bahamas | 13,880 | 316,000 | 30.02 | Independent Former British Colony | 1 |
| Barbados | 430 | 288,000 | 30.96 | Independent Former British Colony | 1 |
| Belize | 22,966 | 328,000 | 22.25 | Independent Former Colony of British Honduras | 1 |
| Bermuda | 54 | 69,000 | 59.25 | British Overseas Territory | 0 |
| Bonaire | 288 | 16,000 | 35.71 | Special Municipality within the Country of the Netherlands | 0 |
| British Virgin Islands | 151 | 31,000 | 23.40 | British Overseas Territory | 0 |
| Cayman Islands | 264 | 53,000 | 41.44 | British Overseas Territory | 0 |
| Cuba | 110,860 | 11,075,000 | 13.00 | Communist State Former Spanish Colony | 1 |
| Curacao | 444 | 146,000 | 35.49 | Country within the Kingdom of the Netherlands | 0 |
| Dominica | 751 | 73,000 | 33.75 | Independent Former British Colony | 1 |
| Dominican Republic | 49,000 | 10,089,000 | 16.00 | Independent Former Spanish Colony | 1 |
| French Guyana | 91,000 | 200,000 | 11.55 | Overseas Department of France | 0 |
| Grenada | 344 | 109,000 | 35.08 | Independent Former British Colony | 1 |
| Guadeloupe | 1780 | 440,000 | 11.55 | France Overseas Departments | 0 |
| Guyana | 214,969 | 742,000 | 19.53 | Independent Former British Colony | 1 |
| Haiti | 27,750 | 9,802,000 | 17.60 | Independent Former French Colony | 1 |
| Jamaica | 10,991 | 2,889,000 | 30.28 | Independent Former British Colony | 1 |
| Martinique | 1100 | 436,000 | 11.55 | France Overseas Department | 0 |
| Montserrat | 102 | 5000 | 38.18 | British Overseas Territory | 0 |
| Nevis * | 93 | 12,000 | 33.94 | Island State in the Federation of St. Kitts and Nevis Independent Former British Colony | 1 |
| Puerto Rico | 13,790 | 3,691,000 | 26.00 | Unincorporated Organized Territory of the United States | 0 |
| Saba | 13 | 1800 | 27.79 | Special Municipality of the Netherlands | 0 |
| St. Barthelemy | 22 | 7400 | N.A | France Overseas Collectivity | 0 |
| St. Eustatius | 21 | 3500 | 27.79 | Special Municipality of the Netherlands | 0 |
| St. Kitts * | 186 | 35,000 | 33.94 | Island State in the Federation of St. Kitts and Nevis Independent Former British Colony | 1 |
| St. Lucia | 617 | 17,400 | 24.26 | Independent Former British Colony | 1 |
| St. Maarten | 34 | 41,000 | 27.79 | Special Municipality of the Netherlands | 0 |
| St. Martin | 53 | 37,000 | N.A | France Overseas Collectivity | 0 |
| St. Vincent and the Grenadines | 389 | 103,500 | 30.80 | Independent Former British Colony | 1 |
| Suriname | 163,820 | 560,000 | 7.00 | Independent Former Dutch Colony | 1 |
| Trinidad and Tobago | 5128 | 1,226,000 | 4.53 | Independent Former British Colony | 1 |
| Turks and Caicos | 948 | 46,300 | 21.50 | British Overseas Territory | 0 |
| United States Virgin Islands | 1910 | 105,000 | 29.00 | Organized Unincorporated US Territory | 0 |

Data sources: CIA Factbook, Carilec, USEIA. * Note: The political status of a jurisdiction is 1 if it is an independent former colony; it is 0 if it is a dependent territory. N.A: Not applicable.

**Table 2.** Respondent groups and their ratings of the electricity sector as a leader of environmental performance.

| Stakeholder Groups | N | Descriptive Statistics and Aggregated Responses in Percentages for the 11-Point Likert Scale | | | | | | | | | | | | | |
|---|---|---|---|---|---|---|---|---|---|---|---|---|---|---|---|
| | | n | Mean | SD | 0 | 1 | 2 | 3 | 4 | 5 | 6 | 7 | 8 | 9 | 10 |
| Electricity sector stakeholders: electricity producers, electricity utility providers | 21 | 19 | 6.63 | 2.19 | 0% | 0% | 10.53% | 0% | 0% | 21.05% | 0% | 36.84% | 10.53% | 15.79% | 5.26% |
| Government representatives and policymakers | 48 | 42 | 5.64 | 2.05 | 4.76% | 2.38% | 2.38% | 2.38% | 2.38% | 28.57% | 19.05% | 23.81% | 11.90% | 2.38% | 0% |
| International development organizations | 16 | 12 | 3.58 | 2.57 | 8.33% | 8.33% | 25.00% | 16.67% | 8.33% | 16.67% | 0% | 8.33% | 0% | 8.33% | 0% |
| Other primary energy stakeholders: traditional oil and gas companies, renewable energy companies | 20 | 16 | 5.25 | 2.11 | 0% | 0% | 6.25% | 18.75% | 12.5% | 25.00% | 0% | 31.25% | 0% | 0% | 6.25% |
| Primary non-energy stakeholders: manufacturing industries, local agencies of international development organizations | 17 | 15 | 4.33 | 2.26 | 0% | 13.33% | 6.67% | 20.00% | 13.33% | 13.33% | 20.00% | 6.67% | 0% | 6.67% | 0% |
| Secondary stakeholders: locally based domestic and international NGOs and locally based domestic research organizations including universities | 18 | 15 | 5.13 | 2.17 | 0% | 6.67% | 6.67% | 6.67% | 13.33% | 26.67% | 13.33% | 13.33% | 6.67% | 6.67% | 0% |

Notes: N is the total number of persons surveyed under each category. n is the number of respondents to the question on the electricity sector as a leader of environmental performance. % refers to the percentage of respondents' rating of the electricity sector as a leader of environmental performance for each stakeholder group.

**Table 3.** Factor loading: Nature of interactions with international development organizations.

| Scheme 2. | Perceived International Development Organizations' Attitudes and Interests toward the Electricity Sector and the SIJs |
|---|---|
| Q2 (Understand my country) | 0.806 |
| Q3 (Understand my company) | 0.731 |
| Q4 (Provide support) | 0.653 |
| Q5 (Interest in my company) | 0.614 |
| Q1 (Fair manner) | 0.0537 |
| Q6 (Skills learned) | |
| Q7 (Knowledge learned) | |

Notes: Extraction method: exploratory factor analysis. Rotation method: Varimax with Kaiser normalization.

(3) Governments: Local governments' attitudes. To measure the role of the local governments in the Caribbean SIJs' energy sector, there were six questions in the questionnaire on the stakeholders' perceptions of the general attitudes of governments on economic development, environmental concerns, and sustainable development. Through a second exploratory factor analysis of their answers, one major factor emerged, namely the perceived government support for sustainable development. The specific factor loadings are shown in Table 4.

**Table 4.** Factor loadings: General attitudes of government.

| Survey Question | Perceived Government Support for Sustainable Development |
|---|---|
| Q10 (Climate change) | 0.741 |
| Q11 (Technology) | 0.686 |
| Q9 (Sustainable development) | 0.449 |
| Q13 (Needs of poor community) | |
| Q8 (Environmental concerns) | |
| Q12 (Burden on the economy) | |

Notes: Extraction method: exploratory factor analysis. Rotation method: Varimax with Kaiser normalization.

(4) Governments: Involvement of electricity sector stakeholders in policymaking. Respondents (i.e., all stakeholders) were asked to rate the statement, "The government often changes energy policy direction following recommendations made by electricity sector stakeholders" to measure the perceived government involvement of the electricity sector actors in energy policymaking.

(5) Governments: Involvement of other primary energy stakeholders in policymaking. Respondents (i.e., all stakeholders) were asked to rate the statement, "The government often changes energy policy direction following recommendations made by other energy sector actors" to measure the perceived government involvement of the other energy stakeholders in energy policymaking.

Moderator. To capture any nuances or heterogeneity that may exist at the level of stakeholders' perceptions across the 34 island economies in the Caribbean, we used their political jurisdictions as a moderator. The political jurisdiction variable is a dummy variable where stakeholders from independent SIJs were assigned a value of 1 and those from dependent SIJs were assigned a value of 0 (Table 1).

### 2.2.3. Model Specification

We tested the impact of local governments and international development organizations on our dependent variable—the perceived positioning of the electricity sector as a leader in environmental performance—by using a bootstrapped OLS regression model. OLS regression is the most common estimation technique to estimate the relationships between one or more independent quantitative variables and a dependent variable [44].

The bootstrapping method handles statistical issues associated with small samples and increases the reliability of the results [45]. We also tested the impact of the political jurisdiction as an interactive term with our independent variables. The measurement model was as follows:

$$Y_i = \alpha_i + \beta_1 X_i + \beta_2 X * M + \varepsilon$$

$Y_i$ is the dependent variable 'environmental leader', $\alpha$ is the constant. $X_i$ is the set of independent variables, representing the two leading actors in the Caribbean SIJs (local governments and international development organizations). $X * M$ is our moderating effect between political jurisdiction and specific independent variables, $\varepsilon$ is the error term.

## 3. Results

The detailed descriptive statistics for the variables are presented in Table 5 and the Pearson correlation coefficients are shown in Table 6. We ran seven regression models and we present the results in Table 7.

**Table 5.** Descriptive statistics of the variables.

| Variable Name | Description | Type | Observations | Mean | St. Dev. |
|---|---|---|---|---|---|
| **Dependent variable** | | | | | |
| Environmental leader | Electricity sector actors are perceived as leaders in environmental performance | Numerical (11-Likert Scale) | 119 | 5.31 | 2.30 |
| **Independent variables** | | | | | |
| **Local governments** | | | | | |
| Factor: Government support for sustainable development | Government's supportive attitudes towards climate change, renewable technologies, and sustainable development | Factor loading | 132 | 0 | 0.84 |
| Government's involvement of the electricity sector stakeholders in policymaking | Government's willingness to change policies based on recommendations from the electricity sector stakeholders | Dummy: 1—Government is open response, 0—otherwise | 140 | 0.19 | 0.39 |
| Government's involvement of other primary energy stakeholders in policymaking | Government's willingness to change policies based on recommendations from other energy stakeholders | Dummy: 1—Government is open response, 0—otherwise | 140 | 0.44 | 0.50 |
| **International development organizations** | | | | | |
| Number of interactions with international development organizations | Number of international development organizations (Organization of American States, Inter-American Development Bank, World Bank, United Nations, European Union) in the country | Numerical (number of international development organizations) | 140 | 2.61 | 2.31 |
| Factor: International development organizations' attitudes | International development organizations' attitudes and interests toward the electricity sector and the SIJs | Factor loading | 115 | 0 | 0.89 |
| **Moderator** | | | | | |
| Political jurisdiction | Independent SIJs | Dummy: 1 if independent economy, 0 if dependent territory | 140 | 0.75 | 0.43 |

Model 1 reports the results on the stakeholders' perceptions of the positioning of the electricity sector as a leader in environmental performance as impacted by governments and international development organizations. The mean variance inflation factor (VIF) for

the model is 1.15 with no single VIF over 2, suggesting multicollinearity is not a significant concern [46]. The results suggest that governments' involvement of other (non-electricity) primary energy stakeholders has a positive effect on the perception of the electricity sector as an environmental leader. Thus, our Hypothesis 1a is supported.

Models 2 to 7 include the results for the moderating variable, i.e., the political jurisdiction of the different Caribbean SIJs. As shown in Models 3 to 6, there is no significant effect on the perception of the electricity sector as an environmental leader from the interaction between the political jurisdiction of the SIJs and the other independent variables.

In Model 7, there is a positive moderating effect of political jurisdiction status on the relationship between a government's involvement of other energy stakeholders and the perception of the electricity sector as an environmental leader ($r = 2.583$ *, $p < 0.05$). Thus, Hypothesis 2a is supported. Model 7 reveals that in independent Caribbean SIJs, when the national governments involve other primary non-electricity energy stakeholders (renewable energy companies and traditional oil and gas companies) in energy policymaking, the electricity sector actors are better viewed as leaders in environmental performance. In other words, national governments and the other primary energy stakeholders are perceived as the influential actors in energy policymaking, while in dependent Caribbean SIJs, no such dynamics regarding influential actors exist. Figure 2 illustrates the moderation effects of political jurisdiction on the relationship between governments' involvement of other energy stakeholders in policymaking and the leadership of the electricity sector in terms of environmental performance.

**Table 6.** Pearson correlation table.

|  | 1 | 2 | 3 | 4 | 5 | 6 | 7 |
|---|---|---|---|---|---|---|---|
| 1. Environmental leader | 1.00 |  |  |  |  |  |  |
| 2. International development organizations' attitudes | 0.23 * | 1.00 |  |  |  |  |  |
| 3. Government's support for sustainable development | 0.19 * | 0.24 * | 1.00 |  |  |  |  |
| 4. Government's involvement of electricity sector stakeholders | 0.02 | 0.05 | 0.09 | 1.00 |  |  |  |
| 5. Government's involvement of other energy stakeholders | 0.32 *** | 0.37 *** | 0.23 ** | 0.17 * | 1.00 |  |  |
| 6. Political jurisdiction | −0.03 | −0.17 + | 0.01 | −0.06 | −0.05 | 1.00 |  |
| 7. Number of interactions with international development organizations | −0.04 | −0.17 + | −0.02 | −0.07 | 0.08 | 0.35 *** | 1.00 |

Note: Pearson correlations were reported and two-tailed t-tests were performed. +: $p < 0.1$; *: $p < 0.05$; **: $p < 0.01$; ***: $p < 0.001$.

**Table 7.** Bootstrapped OLS regression models.

| | Environmental Leader | | | | | | |
|---|---|---|---|---|---|---|---|
|  | Model 1 | Model 2 | Model 3 | Model 4 | Model 5 | Model 6 | Model 7 |
| X1: International organizations: Interactions | −0.052 (0.128) | −0.064 (0.128) | 0.080 (0.421) | −0.064 (0.128) | −0.067 (0.128) | −0.068 (0.129) | −0.070 (0.127) |
| X2: International organizations: Attitudes | 0.221 (0.346) | 0.240 (0.343) | 0.251 (0.351) | 0.228 (0.734) | 0.241 (0.349) | 0.241 (0.344) | 0.164 (0.339) |
| X3: Governments: Support for sustainable development | 0.249 (0.309) | 0.238 (0.317) | 0.241 (0.320) | 0.238 (0.317) | −0.094 (0.826) | 0.239 (0.322) | 0.335 (0.315) |
| X4: Governments: Involvement of electricity sector stakeholder in policymaking | −0.252 (0.499) | −0.254 (0.500) | −0.223 (0.513) | −0.252 (0.509) | −0.243 (0.503) | −0.702 (1.557) | −0.017 (0.491) |

**Table 7.** *Cont.*

| | Environmental Leader | | | | | | |
|---|---|---|---|---|---|---|---|
| X5: Governments: Involvement of other primary energy stakeholders in policymaking | 1.440 ** (0.541) | 1.427 ** (0.540) | 1.414 * (0.554) | 1.426 ** (0.525) | 1.462 ** (0.555) | 1.453 ** (0.546) | −0.643 (1.151) |
| Political jurisdiction | | 0.237 (0.640) | 0.584 (1.151) | 0.233 (0.700) | 0.235 (0.660) | 0.160 (0.740) | −0.987 (0.902) |
| Political jurisdiction × X1 | | | −0.166 (0.435) | | | | |
| Political jurisdiction × X2 | | | | 0.015 (0.797) | | | |
| Political jurisdiction × X3 | | | | | 0.397 (0.872) | | |
| Political jurisdiction × X4 | | | | | | 0.552 (1.651) | |
| Political jurisdiction × X5 | | | | | | | 2.583 * (1.137) |
| Intercept | 4.761 *** (0.563) | 4.617 *** (0.750) | 4.347 *** (1.098) | 4.623 *** (0.792) | 4.602 *** (0.777) | 4.679 *** (0.738) | 5.555 *** (0.894) |
| Number of observations | 101 | 101 | 101 | 101 | 101 | 101 | 101 |
| Wild χ2 (df) | 17.09 (5) ** | 17.77 (6) ** | 18.38 (7) ** | 17.97 (7) * | 18.37 (7) ** | 17.72 (7) * | 26.67 (7) *** |
| bootstrap replications | 500 | 500 | 500 | 500 | 500 | 500 | 500 |
| $R^2$ | 0.1411 | 0.1425 | 0.1447 | 0.1425 | 0.1452 | 0.1436 | 0.1881 |
| Adjusted $R^2$ | 0.0958 | 0.0877 | 0.0803 | 0.0779 | 0.0808 | 0.0791 | 0.1271 |

Notes: Standard errors reported in parentheses. *: $p < 0.05$; **: $p < 0.01$; ***: $p < 0.001$. All the regressions were bootstrapped 500 times.

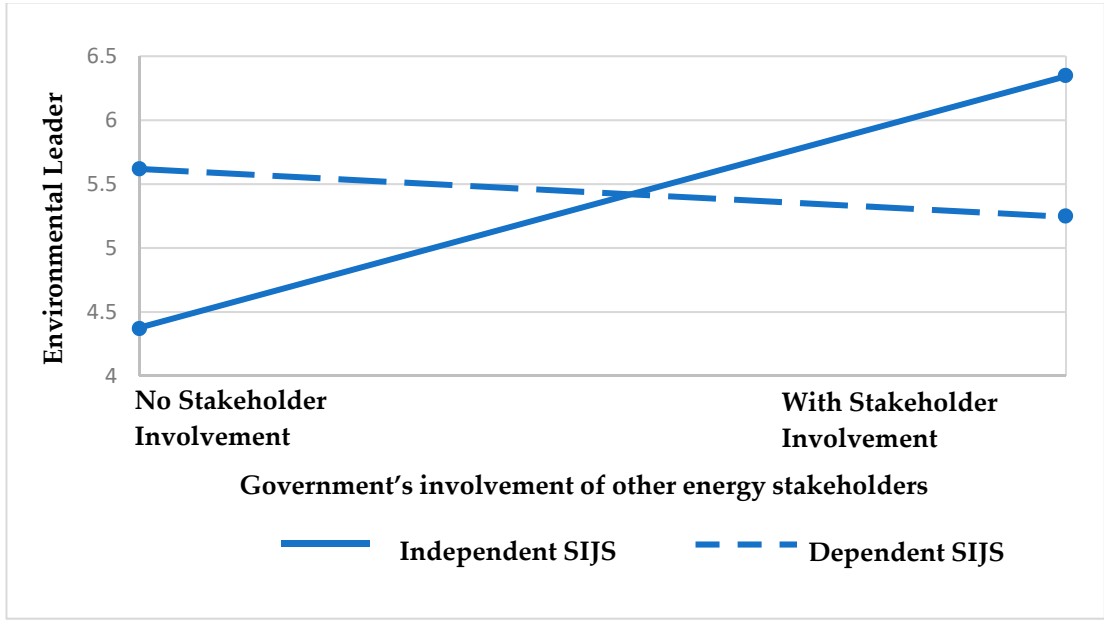

**Figure 2.** The moderation effect of political jurisdiction.

## 4. Discussion

Our study confirms that stakeholders' perceptions across the Caribbean political jurisdictions is heterogenous in energy policymaking and environmental sustainability when political jurisdictions differ. This implies that aiming for a coherent regional 'one size fits all' strategy is challenging to achieve for these 34 economies because they are likely heterogenous in terms of their internal structures as well. Since stakeholders' perceptions

are an important basis for the implementation of sustainable policies and strategies [47], energy policymaking should account for its differences across different Caribbean SIJs. Furthermore, our findings imply that national governments must also tailor their resource allocation and development strategies according to these systemic differences.

Our study also shows that in the case of the independent SIJs only, when national governments involve other primary energy stakeholders (renewable energy companies and traditional oil and gas companies) in energy policymaking, the electricity sector actors are better viewed as leaders in environmental performance (confirming our Hypotheses H1a and H2a). More specifically, the significant regression result in X5—government involvement of other primary energy stakeholders in policymaking—and the insignificant regression result in X4—government involvement of electricity sector stakeholders in policymaking—indicate that the stakeholders' perceptions are rather denunciating in nature. This implies that when energy incumbents are perceived as being in situations of conflicting interests, their perceived exclusion from energy policymaking seems to be an incentive for them to perform better environmentally. Likewise, the significant moderating effect between political jurisdiction and X5—government involvement of other primary energy stakeholders in policymaking—and the insignificant moderating effect of X4—government involvement of electricity sector stakeholders—show that the stakeholders' perceptions of the independent SIJs, aside from being denunciating in nature, also indicate a malaise which emanates from perceived conflicts of interest in the electricity sector, and which can only be counteracted by exclusion.

Our study and results therefore add to the emerging literature on the conflicts of interest between incumbent coal and oil companies and the emerging low-carbon sectors and technologies [43], and explore its implications for energy policymaking, environmental sustainability, as well as the overarching sustainable development strategy of SIJs. Our study is also informative for other developing economies, since the plethora of political jurisdictions within a small geographical distance in the Caribbean region serves as an ideal setting for understanding the multiplicity of institutional environments across economies as related to energy [48]. Indeed, MacArthur and Wilson [48] affirm that for every political jurisdiction in the developing world, an analogous political jurisdiction can be found in the Caribbean. Thus, the institutional diversity in the region can be used as a basis for building hypotheses on how various phenomena may operate across the 'loungers' and 'laggards' [49] of the developing world (i.e., developing economies that are trying to catch up and close the gap with economies that are the followers and leaders in innovating and implementing appropriate national policies for achieving sustainable development). Some developing economies are loungers and laggards in sustainable development because of systemic failures at the national level. These systemic failures include the lack of technology push, the lack of market pull instruments, and the lack of required resources to achieve eco-innovations and implement national policies for sustainable development. They therefore lag behind other economies in their achievement of sustainable development. Loungers are one step ahead of laggards, with a marginally stronger national system of innovation and they are therefore able to slowly catch up with eco-innovation approaches and slowly implement the required national policies for achieving sustainable development. These two categories can be contrasted with the follower and leader economies of the world in terms of eco-innovation and sustainable development [49].

Our results provide insight into the entities stakeholders perceive as making or breaking energy policymaking. Thus, the implication for other SIJs and developing economies is that the exclusion of incumbent energy actors from energy policymaking could be one approach for achieving environmental sustainability and for implementing and achieving a holistic sustainable development strategy. This may be specifically relevant to loungers and laggards that are so labeled due to their inherent systemic failures.

In a global context where inclusiveness is as important a mandate as sustainability, our findings indicate that within fragile systemic contexts, the contentious approach of excluding extant powerful groups or incumbent energy actors in energy policymaking

is perceived by other stakeholders as the way to move the environmental sustainability mandate forward. The carrot and stick analogy has been tirelessly used within the domain of policymaking, and our findings indicate that in order to achieve radical change, contentious measures—sticks—are perceived as being more effective than carrots within certain economies. Thus, our findings imply an alternative stance to the notion that at the government level, including stakeholders in policymaking processes provides marginal groups with the opportunity to empower their voices [18]. Our findings indicate that unlike marginal groups, extant powerful groups should either not be given a voice in energy policymaking, or their voices should be moderated by national governments to ensure that the former do not engage in purely self-serving agendas and to achieve greater social welfare, such as better environmental performance, as examined in this study.

Even though Innes and Booher [50] state that the exclusion of particular stakeholders from policymaking may cause an agreement to fail due to a perceived lack of legitimacy from the public's standpoint, we empirically show in this study that the public has an opposite perceived standpoint.

Our study also adds insight into the regional energy transition literature. Understanding the Caribbean SIJ stakeholders' perceptions of the electricity sector's contributions towards environmental sustainability is one of the many necessary factors for ensuring energy security and for facilitating the energy transition processes within the region. The latter is particularly valuable since the Caribbean region is the second most environmentally hazard-prone region in the world in terms of natural disasters, climate change, loss of biodiversity, pollution, and depleted freshwater [10].

## 5. Conclusions

While our results suggest the exclusion of incumbent energy actors, we propose the adoption of a 'mindful' approach for energy policymaking and sustainability. We advance that policymakers should be mindful of the potential biases that non-renewable or incumbent energy actors may have due to their substantial financial and survival stake in any energy transition. In our study, the role of the incumbent actors within the electricity sector is pertinent as they understand the local particulars of the jurisdiction's electricity system, but their potential conflicts of interest should be 'on the table' in transition discussions. Putting potential conflicts of interest 'on the table' is good governance [51]. However, very much like the carrot and stick analogy, this latter context is reminiscent of the chicken and egg analogy. In other words, in the systemic contexts that we are analyzing, described by the OECD [10] as including among other factors incomplete or inadequate governance frameworks, the issue of what comes first—good governance or declaration of conflicts of interest—remains a conundrum.

It can be argued that exclusion from energy policymaking may deter domestic and international investment in the renewable energy sectors. As a countermeasure, we propose that local governments focus their efforts on providing potential investors with specific location advantages (e.g., good infrastructure and support services, social capital, and technological, managerial, relational, and other created assets) that motivate them to shift from non-renewable to renewable sources of energy.

With regards to practitioners, especially those from incumbent and powerful energy firms, the implication is that there are advantages to stepping back and not engaging in policymaking and to engaging in positive environmental performance irrespective of the stakes at hand. Such an approach by practitioners, in similar developing economy contexts, may enrich social trust by improving how their firms are perceived by other stakeholders and society in general.

We make several contributions by exploring the Caribbean SIJs' heterogeneity in terms of political jurisdictions (and the associated economic vulnerability). The first contribution we make is in terms of integrating stakeholder heterogeneity in the discussion on Caribbean SIJs, but using the prevalent indicator of heterogeneity as a moderator. Our study, therefore, begins to address the pertinence of politics and power within the Caribbean context

and how stakeholders view themselves within this dynamic. The second contribution is that by further exploring the conflicts of interest between the electricity sector and other stakeholders, we explained the power of incumbent energy actors and the role of all local stakeholders in shaping the energy transition for the Caribbean SIJs. Third, we discussed local stakeholders' ability and scope for collaborative decision-making in such a complex political–power dynamic and as well as their perceptions of this dynamic.

A further investigation of this complex dynamic is a fruitful area for future research. Other factors such as levels of economic dependence/independence, economic vulnerability/strength, levels of foreign debt, heterogeneity in energy policy, the politics of debt relief, colonial and postcolonial histories, and more, in shaping the trajectories of energy development are pertinent for the current and future trajectories of energy transition within the region. Thus, we recommend that future researchers consider these endogenous and exogenous factors in their analysis of the Caribbean SIJs' stakeholders, environmental sustainability, and incumbent energy actors. Such future studies would offer some powerful recommendations with respect to shaping locally responsive energy policies and promote energy development trajectories which are cognizant of the larger complex political–power dynamic at play. Indeed, protocols and agreements such as the Clean Development Mechanism designed by international organizations and regimes may not reap the desired effects when applied in a wholesale manner and without accounting for local context specificities.

Our study's main limitation is the small size of each stakeholder group of respondents, which prevented an analysis of the effect of each stakeholder group on our main dependent variable and in relation to the moderator variable. Future studies could be implemented to investigate each specific stakeholder group's perceptions and the interactions among different stakeholder groups by increasing the sample size across different Caribbean SIJs.

**Author Contributions:** Conceptualization, X.L., J.A.P. and C.D.F.; methodology, X.L., J.A.P. and C.D.F.; software, X.L.; validation, X.L. and C.D.F.; formal analysis, X.L., J.A.P. and C.D.F.; investigation, X.L., J.A.P. and C.D.F.; resources, D.I. and H.V.; data curation, X.L. and C.D.F.; writing—original draft preparation, X.L., J.A.P. and C.D.F.; writing—review and editing, X.L., J.A.P., C.D.F., D.I. and H.V. All authors have read and agreed to the published version of the manuscript.

**Funding:** This research was funded by Suncor Energy Foundation for providing research fundings for data gathering.

**Institutional Review Board Statement:** The study was conducted according to the guidelines of the the University of Calgary and the Tri-Council Policy Statement, and approved by the Research Ethics Board of the University of Calgary (File No: 6439; Date of approval: 19 May 2010).

**Informed Consent Statement:** Informed consent was obtained from all subjects involved in the study.

**Acknowledgments:** We thank the three anonymous reviewers whose comments helped improve and clarify this manuscript. We thank Kate Ervine for her friendly review and constructive suggestions.

**Conflicts of Interest:** The authors declare no conflict of interest.

## Appendix A

**Table A1.** Survey Questions.

| Variables | Survey Questions |
|---|---|
| Environmental leader | What is YOUR perception of the electricity sector in your local economy? Please answer by indicating to what extent you agree or disagree with the following statement with respect to the electricity sector. Please 'click' the point on the scale that most accurately represents YOUR level of agreement/disagreement. If you neither agree nor disagree, leave the slider pointer on 'neutral'. (11 Likert scale, 0—strongly disagree, 5—neutral, 10—strongly agree) The electricity sector is a leader in my local business environment and economy in terms of environmental performance. |

**Table A1.** *Cont.*

| Variables | Survey Questions |
| --- | --- |
| The number of foreign development organizations | Identify which, if any, of the following agencies of foreign development organizations you have interacted with during the course of your professional career. Note: If you have worked with more than one of these, click all that apply.<br>Organization of American States<br>Inter-American Development Bank<br>World Bank<br>United Nations<br>European Union<br>Others, please specify |
| The nature of interactions with international development organizations | Please 'click' on the point of the scale that most accurately represents YOUR level of agreement/disagreement with the following statements. If you neither agree nor disagree, leave the slider on 'neutral'.<br>(11 Likert Scale, 0—strongly disagree, 5—neutral, 10—strongly agree)<br>Q1. The representative agencies that I have interacted with have always dealt with me in a fair manner.<br>Q2. The representative agencies I have interacted with have a good grasp of my specific country's situation.<br>Q3. The representative agencies I have interacted with have a good grasp of my organization's objectives.<br>Q4. The representative agencies I have interacted with have provided what I needed from them.<br>Q5. The representative agencies I have worked with have a keen interest in my organization's future requirements.<br>Q6. The representative agencies I have worked with have significantly improved the skills of persons working in my organization.<br>Q7. I have acquired significant knowledge through my interaction with their representative agencies. |
| General government attitudes towards economic development, environmental concerns and sustainable development | In the section below, please 'click' on the appropriate point on the scale to indicate the extent to which you agree/disagree that the following statements reflect YOUR GOVERNMENT'S general attitude. Do NOT provide your own level of agreement with the statements. If you neither agree nor disagree that the statements reflect YOUR GOVERNMENT'S opinion, leave the slider at 'neutral'.<br>(11 Likert scale, 0—strongly disagree, 5—neutral, 10—strongly agree)<br>Q8. Environmental concerns should not be sacrificed for economic growth.<br>Q9. Sustainable development is critical to national development.<br>Q10. It is important to address the issue of climate change.<br>Q11. The development of renewable energy technologies will offer opportunities for local businesses.<br>Q12. The development of renewable energies will place a significant burden on the economy.<br>Q13. Renewable energy development must not take place without ensuring that the needs of poor communities are also addressed. |
| Government's involvement of the electricity sector stakeholders in policymaking | Please 'click' on the appropriate point of the scale to indicate the extent to which you agree/disagree with the statement as it relates to the government ministry/ ministries responsible for the energy sector. If you neither agree nor disagree with the statement, leave the slider at 'neutral'.<br>(11 Likert scale, 0—strongly disagree, 5—neutral, 10—strongly agree)<br>The government often changes energy policy direction following recommendations made by electricity sector stakeholders. |
| Government's involvement of other primary energy stakeholders' in policymaking | Please 'click' on the appropriate point of the scale to indicate the extent to which you agree/disagree with the statement as it relates to the government ministry/ministries responsible for energy. If you neither agree nor disagree with the statement, leave the slider at 'neutral'.<br>(11 Likert scale, 0—strongly disagree, 5—neutral, 10—strongly agree)<br>The government often changes energy policy direction following recommendations made by other energy stakeholders. |

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
