# Peer review of "Who Makes or Breaks Energy Policymaking in the Caribbean Small Island Jurisdictions? A Study of Stakeholders’ Perceptions"

_sustainability, doi:10.3390/su14031902_

Round 1
Reviewer 1 Report
The study is interesting, however, it cannot be accepted in its current form. The following could be considered while revising the paper:
- The introduction is too long. A reader can easily get lost. The authors may review it and make it more coherent.
- Section 2 is important but the way it is presented may lead to confusion or misunderstanding. This part also can be revised to improve its clarity.
Author Response
We thank the Editor and Editorial team of Sustainability for the opportunity to submit a revised version of our manuscript.
We also thank all reviewers for the positive feedback and constructive suggestions. We made careful revision to our manuscript based on the reviewers’ recommendations. We hope that the reviewers find this revised manuscript satisfactory. Below we outline, point-by-point, the revisions that we have made in response to the reviewers’ comments.
- We revised the Introduction and Section 2 of the manuscript as per Reviewer 1 and 2’s suggestions.
- We added the proposed reference in the manuscript, following Reviewer 2’ suggestion.
- Following Reviewer 2 and 3’s suggestions, we used the journal submission template to reorganize this revised manuscript and modified the page layout to meet the guidelines and conventions of the journal.
- As per Reviewer 3’s comments, we added discussions to support our adopted OLS regression method.
- Following Reviewer 3’s suggestion, we revised the conclusion section and shortened the paper.
Our detailed responses to the comments from Reviewer 1 are below. Please feel free to contact us if you need any additional information.
Kind regards,
The Authors
Reviewer 1
Comments to the Author
The study is interesting; however, it cannot be accepted in its current form. The following could be considered while revising the paper:
- The introduction is too long. A reader can easily get lost. The authors may review it and make it more coherent.
Response:
Thank you very much for your comments. We rewrote the introduction section to make it more focused. We began the introduction by outlining the scope of our research. We highlighted the research gap, proposed our research questions, presented our hypotheses and our contributions.
- Section 2 is important but the way it is presented may lead to confusion or misunderstanding. This part also can be revised to improve its clarity.
Response:
We revised Section 2 significantly to include “Materials and methods” according to the requirement of the journal.
In “Materials”, we conducted literature review and proposed our hypotheses. The literature review includes three major components:
- The conceptual framework of the study.
- The importance of stakeholder perception and stakeholder inclusion for sustainable development, which build the theoretical foundation for our arguments.
- The importance of governments and international organizations in achieving sustainability, which forms the basis for the first set of hypotheses.
- The moderating role played by political jurisdictions of Caribbean SIJs, which forms the basis for the second set of hypotheses.
In “Methods”, we outlined the research background and data collection. Then we explained our dependent, independent, and moderator variables. We end this subsection of the revised paper by providing the measurement model.
All changes are tracked in the revised manuscript. We also provide a clean version of our revised manuscript.
Reviewer 2 Report
The manuscript attempt to examine inter-small island ju-risdictions (SIJs) heterogeneity from the social construct perspective of stakeholders’ perception and within the context of environmental sustainability and energy policy-making. The paper contains new and significant information adequate to justify publication. It demonstrates an adequate understanding of the relevant literature in the field and cites an appropriate range of literature sources. Literature is extensive and up to date. The paper's argument is built on an appropriate base of theory and concepts. The methods are employed appropriately and are described precisely in the manuscript. In my opinion, the article should consider papers like for example Maris and Flouros (2021) The green deal, national energy and climate plans in Europe: Member States’ compliance and strategies. Administrative Sciences, 11(3), 75, in order to enrich the framework of responses, strategies and compliance.
The author/s must be sure that the manuscript format meets the high journal standards. In my opinion, they must modify the page layout to meet the guidelines and conventions of the journal.
I am thankful and honored for the opportunity to read your manuscript.
It is a good, interesting, and inspiring article!
Best wishes!
Author Response
We thank the Editor and Editorial team of Sustainability for the opportunity to submit a revised version of our manuscript.
We also thank all reviewers for the positive feedback and constructive suggestions. We made careful revision to our manuscript based on the reviewers’ recommendations. We hope that the reviewers find this revised manuscript satisfactory. Below we outline, point-by-point, the revisions that we have made in response to the reviewers’ comments.
- We revised the Introduction and Section 2 of the manuscript as per Reviewer 1 and 2’s suggestions.
- We added the proposed reference in the manuscript, following Reviewer 2’ suggestion.
- Following Reviewer 2 and 3’s suggestions, we used the journal submission template to reorganize this revised manuscript and modified the page layout to meet the guidelines and conventions of the journal.
- As per Reviewer 3’s comments, we added discussions to support our adopted OLS regression method.
- Following Reviewer 3’s suggestion, we revised the conclusion section and shortened the paper.
Our detailed responses to the comments from Reviewer 2 are below. Please feel free to contact us if you need any additional information.
Kind regards,
The Authors
Reviewer 2
Comments to the Author
The manuscript attempt to examine inter-small island jurisdictions (SIJs) heterogeneity from the social construct perspective of stakeholders’ perception and within the context of environmental sustainability and energy policy-making. The paper contains new and significant information adequate to justify publication. It demonstrates an adequate understanding of the relevant literature in the field and cites an appropriate range of literature sources. Literature is extensive and up to date. The paper's argument is built on an appropriate base of theory and concepts. The methods are employed appropriately and are described precisely in the manuscript.
Response:
Thank you very much for your positive feedback on our manuscript.
In my opinion, the article should consider papers like for example Maris and Flouros (2021) The green deal, national energy and climate plans in Europe: Member States’ compliance and strategies. Administrative Sciences, 11(3), 75, in order to enrich the framework of responses, strategies and compliance.
Response:
Indeed, the contribution from Maris and Flouros (2021) helped us enrich the understanding of the importance of policy making and different response rate from different stakeholders with regards to environment and energy policies. Specifically, we added the reference in the discussion of subsection 2.1.3, as follows.
“Furthermore, involving stakeholders in different political settings have a different response pace with regards to environment and energy policies 40. As a result, the different response pace might lead to conflicts of interests among stakeholders. As indicated by Ince et al. 18, there are conflict of interests between the incumbent electricity sector and independent renewable energy producers in the Caribbean SIJs. It is thus necessary to design effective policies to reconcile the conflicting interests of incumbent coal and oil companies with those of the emerging low-carbon sectors and technologies 43.”
The author/s must be sure that the manuscript format meets the high journal standards. In my opinion, they must modify the page layout to meet the guidelines and conventions of the journal.
Response:
In this revised version, we have used the submission template of the journal to prepare manuscript. We made sure it meets the guidelines and conventions of the Sustainability journal.
All changes are tracked in the revised manuscript. We also provide a clean version of our revised manuscript.
I am thankful and honored for the opportunity to read your manuscript. It is a good, interesting, and inspiring article!
Response:
Thank you again for your positive and encouraging comments. We are glad that you found the paper interesting and inspiring.
40 Maris, G.; Flouros, F., The green deal, national energy and climate plans in Europe: Member States’ compliance and strategies. Administrative Sciences 2021, 11 (3), 75.
18 Ince, D.; Vredenburg, H.; Liu, X., Drivers and inhibitors of renewable energy: A qualitative and quantitative study of the Caribbean. Energy Policy 2016, 98, 700-712.
43 Gupta, D.; Ghersi, F.; Vishwanathan, S. S.; Garg, A., Achieving sustainable development in India along low carbon pathways: Macroeconomic assessment. World Development 2019, 123, 104623.
Reviewer 3 Report
The subject of the article is interesting and worth describing. However, the method of implementation is inadequate. In the Introduction, the authors presented an introduction to the subject. The Introduction section is deficient. Research hypotheses should also be provided. They can correspond with research questions. Additionally, the research gap should be clearly marked (before the goal, questions and hypotheses).
The layout of the work is incorrect. In section 1 of the Introduction, the hypotheses and a clear research gap are missing.
Section 3 title is inappropriate (Methodology?). Methodology is the science of scientific research methods. The section should be called Materials and Methods, and it should be section 2. There is a place to organize your research. Provide the scope of the research, the methods used, the sequence of tests. It is not there at the moment. Figure 1 should be transferred to section 2. Materials and Methods.
I have doubts about the representativeness of the results obtained due to the incomplete return of the questionnaires. Additionally, the surveyed group is highly diversified. A mere examination of the data contained in Table 1 shows that these are in many respects incomparable objects. Therefore, it is obvious, even without statistical analyzes, that they will be heterogeneous also in terms of energy policy. It is also difficult to draw any objective conclusions from such a heterogeneous group. Of course, in the statistics we will get a certain result, but whether it will not be wrong. Is Pearson's linear correlation method suitable for this type of analysis?
Section 5. Conclusion is not entirely appropriate. Conclusions should not be so descriptive, but rather synthetic. I do not understand the reference to other literature at this point. This must be done in the Discussion section. The conclusions can be bulleted. You must certainly refer to the hypotheses set at work.
The article is too extensive. It has 44 pages. Redundant parts should be dispensed with. The descriptions and references to the theory are too long. It has to be done synthetically. As a rule, articles in Sustainability are about 20 pages long.
The article was poorly formatted. There is too much space between the lines. The citations should also be different, the reference numbers should be given. Footnotes are also unacceptable. The text is not formatted in several places (it should be justified, e.g. pages 6, 19). The literature list is made wrong. Overall, the Authors did not comply with the requirements of the preparation of the article.
Author Response
We thank the Editor and Editorial team of Sustainability for the opportunity to submit a revised version of our manuscript.
We also thank all reviewers for the positive feedback and constructive suggestions. We made careful revision to our manuscript based on the reviewers’ recommendations. We hope that the reviewers find this revised manuscript satisfactory. Below we outline, point-by-point, the revisions that we have made in response to the reviewers’ comments.
- We revised the Introduction and Section 2 of the manuscript as per Reviewer 1 and 2’s suggestions.
- We added the proposed reference in the manuscript, following Reviewer 2’ suggestion.
- Following Reviewer 2 and 3’s suggestions, we used the journal submission template to reorganize this revised manuscript and modified the page layout to meet the guidelines and conventions of the journal.
- As per Reviewer 3’s comments, we added discussions to support our adopted OLS regression method.
- Following Reviewer 3’s suggestion, we revised the conclusion section and shortened the paper.
Our detailed responses to the comments from Reviewer 3 are below. Please feel free to contact us if you need any additional information.
Kind regards,
The Authors
Reviewer 3
Comments to the Author
The subject of the article is interesting and worth describing.
Response:
Thank you very much for your positive feedback on our manuscript.
However, the method of implementation is inadequate. In the Introduction, the authors presented an introduction to the subject. The Introduction section is deficient. Research hypotheses should also be provided. They can correspond with research questions. Additionally, the research gap should be clearly marked (before the goal, questions and hypotheses).
The layout of the work is incorrect. In section 1 of the Introduction, the hypotheses and a clear research gap are missing.
Response:
We rewrote the introduction section to make it more focused. We began the introduction by outlining the scope of our research. We highlighted the research gap, proposed our research questions, presented our hypotheses and our contributions.
Section 3 title is inappropriate (Methodology?). Methodology is the science of scientific research methods. The section should be called Materials and Methods, and it should be section 2. There is a place to organize your research. Provide the scope of the research, the methods used, the sequence of tests. It is not there at the moment. Figure 1 should be transferred to section 2. Materials and Methods.
Response:
We revised section 2 significantly to include “Materials and methods” according to the requirements of the journal.
In “Materials”, we conducted literature review and proposed our hypotheses. The literature review includes three major components:
- The conceptual framework of the study.
- The importance of stakeholder perception and stakeholder inclusion for sustainable development, which build the theoretical foundation for our arguments.
- The importance of governments and international organizations in achieving sustainability, which forms the basis for the first set of hypotheses.
- The moderating role played by political jurisdictions of Caribbean SIJs, which forms the basis for the second set of hypotheses.
In “Methods”, we outlined the research background and data collection. Then we explained our dependent, independent, and moderator variables. We end this subsection of the revised paper by providing the measurement model.
All changes are tracked in the revised manuscript. We also provide a clean version of our revised manuscript.
I have doubts about the representativeness of the results obtained due to the incomplete return of the questionnaires
Response:
Indeed, the incomplete return of the questionnaires might lead to response bias. We used Qualtrics to manage our survey. As outlined in the Qualtrics website, we followed the suggested steps to monitor the incomplete responses, including setting a time frame for incomplete survey responses, saving survey progress, and identifying incomplete responses in our data set. As stated in the revised data collection section, “we tested for demographic-based non-response bias and there was no evidence of systematic bias”.
We also state in the concluding section of the revised manuscript that the main limitation of the study is indeed the small size of each stakeholder group, and that this prevented us from conducting an analysis of the effect of each stakeholder group on our main dependent variable and in relation to the moderator variable.
Additionally, the surveyed group is highly diversified. A mere examination of the data contained in Table 1 shows that these are in many respects incomparable objects. Therefore, it is obvious, even without statistical analyzes, that they will be heterogeneous also in terms of energy policy. It is also difficult to draw any objective conclusions from such a heterogeneous group. Of course, in the statistics we will get a certain result, but whether it will not be wrong.
Response:
Indeed, the stakeholders considered in our data collection and analysis are heterogeneous and this heterogeneity might moderate our proposed relationships. Nevertheless, most existing studies including international institutions (e.g., UN, World Bank) treat them as homogenous. Only a small budding literature, which we cite, state that they are heterogenous and that this heterogeneity must be explored within the context of energy policymaking and environmental sustainability. We answer this call with this paper. Hence, we test whether in Caribbean SIJs, various stakeholders’ perception contributes to the positioning of the electricity sector as a leader in environmental performance. Our results also suggest that the involvement of governments and international development organizations play a key role. In addition, in this revised manuscript, we focus on the moderating role of political jurisdiction status.
To address your comment, in our revised analysis we acknowledge that the heterogeneity of stakeholders might moderate our results, and in the conclusion section we propose it as one of the future research directions.
“A further investigation of the complex dynamic is a fruitful area for future research. Other factors such as levels of economic dependence/independence, economic vulnerability/strength, levels of foreign debt, heterogeneity in energy policy, the politics of debt relief, colonial and post-colonial histories, and more, in shaping the trajectories of energy development are pertinent for the current and future trajectories of energy transition within the region. Thus, we recommend that future research consider these endogenous and exogenous factors in their analysis of the Caribbean SIJs’ stakeholders, environmental sustainability and incumbent energy actors. Such future studies would offer some powerful recommendations with respect to shaping locally-responsive energy policies and promote energy development trajectories which are cognizant of the larger complex political-power dynamic at play.”
Is Pearson's linear correlation method suitable for this type of analysis?
Response:
We explained our use of bootstrapped OLS regression model to test our proposed hypotheses in the revised manuscript as follows, “OLS regression is the most common estimation technique to estimate the relationships between one or more independent quantitative variables and a dependent variable. The bootstrapping method handles statistical issues associated with small samples and increases the reliability of the results.” We had the Pearson’s correlation table (Table 6) in our manuscript to describe the strength of the correlations, and it is not our main method of analysis.
Section 5. Conclusion is not entirely appropriate. Conclusions should not be so descriptive, but rather synthetic. I do not understand the reference to other literature at this point. This must be done in the Discussion section. The conclusions can be bulleted. You must certainly refer to the hypotheses set at work.
Response:
We revised the conclusion section based on your suggestions. All changes are tracked in the revised document.
The article is too extensive. It has 44 pages. Redundant parts should be dispensed with. The descriptions and references to the theory are too long. It has to be done synthetically. As a rule, articles in Sustainability are about 20 pages long.
The article was poorly formatted. There is too much space between the lines. The citations should also be different, the reference numbers should be given. Footnotes are also unacceptable. The text is not formatted in several places (it should be justified, e.g. pages 6, 19). The literature list is made wrong. Overall, the Authors did not comply with the requirements of the preparation of the article.
Response:
We used the submission template of the journal to prepare our revised manuscript. We made sure it meets the guidelines and conventions of the Sustainability journal. We deleted all footnotes and reformatted the whole text. We also reformatted the references using ACS style. We have shortened the revised paper.
All changes are tracked in the revised manuscript. We also provide a clean version of our revised manuscript, which is 23 pages long.
Round 2
Reviewer 3 Report
The article has been completely revised and most aspects are appropriate. However, the discussion is still weak. This section should reference other test results. Meanwhile, there is no reference there. Alternatively, part of the text that relates to the results can be transferred from the literature review. Part of the discussion needs to be supplemented.
Literature references should not be superscript. Please check any published article on Sustainability.
Author Response
The article has been completely revised and most aspects are appropriate. However, the discussion is still weak. This section should reference other test results.
Response:
As per your recommendation, we have strengthened the discussion section and we have referenced other relevant results. These can be found under tracked changes in the submitted revised version.
Meanwhile, there is no reference there. Alternatively, part of the text that relates to the results can be transferred from the literature review. Part of the discussion needs to be supplemented.
Response:
We moved some paragraphs from literature review to discussion section, this has improved our discussion and analysis of our results. We also added necessary explanations as a supplement to parts of the discussion section. These can also be found under tracked changes in the submitted revised version.
Literature references should not be superscript. Please check any published article on Sustainability.
Response:
Thank you very much for your suggestion. We followed the reference format to reformat the references throughout the text.
